

**Chemical Composition-Dependent Hygroscopic Behavior of Individual Ambient Aerosol**
**Particles Collected at a Coastal Site**
Li Wu[1,2,+], Hyo-Jin Eom[1,3,+], Hanjin Yoo[1,4], Dhrubajyoti Gupta[1], Hye-Rin Cho[1], Pingqing Fu[2], and
Chul-Un Ro[1,4,*]
[1] Department of Chemistry, Inha University, Incheon 22212, Korea
[2] Institute of Surface-Earth System Science, School of Earth System Science, Tianjin University,
Tianjin 300072, China
[3] Air Quality Research Division, National Institute of Environmental Research, Incheon 22689, Korea
[4] Particle Pollution Management Center, Inha University, Incheon 21999, Korea
[*] ***Correspondence to*** Chul-Un Ro ([curo@inha.ac.kr](mailto:curo@inha.ac.kr))
[+]Authors with equal contributions
**Abstract**
This study investigated the hygroscopic behavior of individual ambient aerosol particles
collected at a coastal site of Jeju Island, Korea. The particles' size change along with phase transitions
during humidification and dehydration processes, and their chemical compositions, were determined
by optical microscopy and scanning electron microscopy-energy dispersive X-ray spectroscopy (SEM-
EDX), respectively. Of the 39 particles analyzed, 24 were aged sea-spray aerosols (SSAs) with diverse
mixing ratios of $Cl^-$ and $NO_3^-$.
The ambient SSAs exhibited multiple deliquescence and efflorescence transitions that were
dominantly influenced by NaCl, $NaNO_3$, $MgCl_2$, $Mg(NO_3)_2$ and organic species covering the surface
of the aged SSAs. For Cl-rich SSAs with $X_{(Na, Mg)Cl} > 0.4$, although some particles showed very slow
water uptake at low RHs = ~30%, two major transitions were observed during the humidification
process, firstly at RH = ~63.8%, regardless of their chemical compositions, which is the mutual
deliquescence relative humidity (MDRH), and secondly at RH = 67.5–73.5%, depending on their
chemical compositions, which are the final DRHs. During the dehydration process, the Cl-rich SSAs
showed single-stage efflorescence at RH = 33.0–50.5%, due to simultaneous heterogeneous
crystallization of inorganic salts. For Cl-depleted SSAs with $X_{(Na, Mg)Cl} < 0.4$, two prompt deliquescence



transitions were observed during the humidification process, firstly at MDRH = 63.8 % and secondly
at RH = 65.4–72.9%. The mutual deliquescence transition was more distinguishable for Cl-depleted
SSAs. During the dehydration process, step-wise transitions were observed at efflorescence RHs (ERHs)
= 24.6–46.0% and 17.9–30.5%, depending on their chemical compositions.

Additionally, aged mineral particles showed partial or complete phase changes with varying

RH due to the presence of SSAs and/or $NO_3^-$ species. In contrast, non-reacted mineral and Fe-rich
particles maintained their size during the entire hygroscopic process. The mixture particles of organic
and ammonium sulfate (AS) exhibited lower deliquescence and efflorescence RHs compared to pure
AS salt, highlighting the impact of organic species on the hygroscopic behavior of AS. These findings
emphasize the complexity of atmospheric aerosols and the importance of considering their composition
and mixing state when modeling their hygroscopic behavior and subsequent atmospheric impacts.

**1 Introduction**

Atmospheric aerosols play a significant role in the global climate by directly scattering or

absorbing incoming solar radiation and indirectly serving as cloud condensation nuclei (Pandis et al.,
1995; Haywood and Boucher, 2000). The hygroscopicity of ambient aerosol particles, critically
depending on their compositions, is of vital importance in understanding their properties, including
their effects on aerodynamic performance, cloud-droplet nucleation efficiency, optical properties, and
heterogeneous chemical reactivity with atmospheric gas-phase species. (Ten Brink, 1998; Krueger et
al., 2003; Wang and Martin, 2007; Wu et al., 2020). However, the study of their hygroscopic behavior
is challenging because ambient aerosols typically exist as complex mixtures of several chemical species,
even at the individual particle level, due to multiphase interactions (Krieger et al., 2012; Pöschl and
Shiraiwa, 2015; Schiffer et al., 2018).

Sea-spray aerosols (SSAs) are a significant component comprising 25-60% of atmospheric

particulate matter mass (Finlayson-Pitts and Pitts, 2000; Song et al., 2022). Understanding the
hygroscopic properties of SSAs is essential for study on aerosol-cloud interactions and global climate
(Schill et al., 2015; Zieger et al., 2017), which however, is still defective owing to their complex
chemical compositions (Meskhidze et al., 2013; Xu et al., 2020; Cochran et al., 2017). Nascent SSAs
are formed when bubbles burst at the sea surface, generating both submicron and supermicron SSAs
from film and/or jet drops (Quinn et al., 2015; Wang et al., 2017). The primary inorganic constituents
of nascent SSAs are $Na^+$, $Cl^-$, and $Mg^{2+}$, followed by $SO_4^{2-}$, $Ca^{2+}$, $K^+$, and other minor compositions



(Seinfeld and Pandis, 2006). Submicron nascent SSAs contain more organic species and fewer
inorganic salts than supermicron ones (Ault et al., 2013; Prather et al., 2013; Wang et al., 2015). In
pristine marine environments, the organics in SSAs mostly originate from phytoplankton activities in
the sea, while in polluted marine environments, non-biodegradable surfactants from anthropogenic
waste run-offs to the sea are supposed to be added (Cochran et al., 2016; Forestieri et al., 2016).
Reactions of SSAs with various atmospheric species, such as $NO_x/HNO_3$, $SO_2/H_2SO_4$, and $CH_3SO_3H$,
within minutes to hours of residence in air further increase the complexity of the chemical compositions
(ten Brink, 1998; Saul et al., 2006; Liu et al., 2007), leading to partially or fully reacted (or aged) SSAs
after Cl depletion (Pósfai et al., 1995; Gard et al., 1998; Laskin et al., 2012; Ault et al., 2014; Wu et al.,
2020). The further reactive uptake of $N_2O_5$ was also reported to be dependent on the chloride to nitrate
ratio of the reacted SSAs and their phases (Ryder et al., 2014). In addition, SSAs interact with volatile
organic carbons (VOCs), secondary organic aerosols (SOAs), etc., in the marine boundary layer (Su et
al., 2022). The presence of primary and secondary organics, biogenic species, sea-salt sulfates (ss-
$SO_4^{2-}$), non-sea-salt sulfates (nss-$SO_4^{2-}$), etc., adds greater complexity to the interdependence of
hygroscopic behavior and heterogeneous reactions in ambient SSAs (Keene et al., 2007; O'Dowd and
de Leeuw, 2007; Ault et al., 2013; Beardsley et al., 2013; Prather et al., 2013).

Many studies have investigated the hygroscopic behavior of both airborne and laboratory-

generated SSAs. It is generally accepted that sea-salt-containing particles result in higher hygroscopic
factors in supermicron particles (Atkinson et al., 2015; Herich et al., 2009). Some single particle
measurements have been reported on ambient fine and coarse mode SSAs that are dominated by
inorganic salt species. For example, environmental transmission electron microscopy was used to
measure the deliquescence and efflorescence relative humidities (DRHs and ERHs) of NaCl-bearing
aerosols, sulfate/chloride containing SSAs, Mg-rich particles, etc., collected from clean and polluted
environments (Wise et al., 2007; Semeniuk et al., 2007). It was found that NaCl moiety in
sulfate/chloride containing SSAs underwent deliquescence at ~75% RH with the sulfate-bearing phases
remaining insoluble, which is similar to the DRH of pure NaCl aerosols, whereas the DRH of the NaCl
moiety was lowered in the presence of soluble compositions like $NaNO_3$. In a follow-up study, the
DRHs and ERHs of laboratory-generated and ambient SSA particles were found to be consistent (Wise
et al., 2009). Similar observations for marine aerosols with insoluble sulfate moieties and a highly
hygroscopic NaCl-moiety were also reported (Freney et al., 2010a). In-situ Raman spectrometry was
used to probe the phase transitions of SSA droplets (80–100 μm) nebulized from sea-water, which



revealed that CaSO$_4$·0.5H$_2$O solidified at RH > 90%, followed by crystallizations of NaCl and
KMgCl$_3$·6H$_2$O at RH = ~55% and ~44%, respectively (Xiao et al., (2008). Optical microscopy
combined with low-$Z$ particle energy-dispersive electron probe X-ray microanalysis (low-$Z$ particle
EPMA) was used to determine 2-D growth factors, phase transition RHs, and chemical compositions
in ambient aerosols, including nascent and reacted/aged SSAs (Ahn et al., 2010). However, the
relationship between hygroscopic properties and the evolving chemical compositions and mixing states
of ambient SSAs remains unclear.

Laboratory-generated inorganic salt particles have been utilized as surrogates to understand and

parameterize the complex hygroscopic properties of SSAs for climate models. Since NaCl constitutes
approximately 80% of nascent SSAs by mass, the hygroscopic behavior of pure NaCl particles has been
extensively studied for parameterizing the thermodynamic and optical properties and cloud activation
efficiency of ambient SSAs (Tang et al., 1997; Niedermeier et al., 2008). However, the hygroscopic
growth factors of ambient or laboratory-generated SSAs are reported to be different from those of pure
NaCl, possibly due to the presence of hydrates such as MgCl$_2$·6H$_2$O, organic substances, or other
impurities (Ahn et al., 2010; Schindelholz et al., 2014; Zieger et al., 2017; Rosati et al., 2021; Guo et
al., 2019; Kong et al., 2018). Consequently, the hygroscopic properties of multicomponent systems
such as mixed cation chlorides (Ge et al., 1996, 1998; Chan et al., 2000; Li et al., 2014b; Gupta et al.,
2015a), sodium salts of mixed anions (Gupta et al., 2015b; Chang and Lee, 2002; Freney et al., 2010b;
Chan et al., 1997), and other mixture systems such as NaCl-MgSO$_4$ (Woods et al., 2010), NaCl-CaSO$_4$
(Freney et al., 2010b), NaCl-(NH$_4$)$_2$SO$_4$ (Tobon et al., 2021) are of special relevance, which can serve
as surrogates for ambient or reacted SSAs.

As discussed in detail elsewhere (Li et al., 2014b; Gupta et al., 2015b), equilibrium

thermodynamics state that binary mixture systems such as NaCl-KCl, NaCl-MgCl$_2$, and NaCl-NaNO$_3$
exhibit multi-stage deliquescence/efflorescence transitions at mutual DRHs/ERHs (MDRHs/MERHs)
due to the dissolution/crystallization of eutonic compositions, which are independent of the initial
mixing ratios, and at specific DRHs/ERHs due to the richer salt moiety, respectively. Thermodynamic
models,        such        as        the        Extended        Atmospheric        Inorganics        Model        (E-AIM)
(http://www.aim.env.uea.ac.uk/aim/aim.php; (Ansari and Pandis, 1999; Carslaw et al., 1995; Clegg et
al., 1998a, b; Wexler and Clegg, 2002) and the Aerosol Inorganic-Organic Mixtures Functional groups
Activity Coefficients (AIOMFAC) (http://www.aiomfac.caltech.edu; Zuend et al., 2008, 2011), can
predict MDRHs and DRHs for multicomponent mixture systems. However, as efflorescence is a kinetic



or rate-driven process, no general theoretical model can predict the efflorescence of single or
multicomponent aerosol particles, and thus the best way is experimental observation (Seinfeld and
Pandis, 2006; Cohen et al., 1987; Martin, 2000). Previous modeling and field studies have attributed
the reduction in hygroscopic growth of SSAs to the organic fractions (Ming and Russell, 2001; Vaishya
et al., 2013; Zhang et al., 2014), whereas recent measurements suggest that organic species have an
insignificant influence (Nguyen et al., 2017). Therefore, establishing a systematic correlation between
the chemical compositions and hygroscopic behavior of ambient SSAs vis-à-vis the multicomponent
inorganic surrogates is a priority towards understanding the hygroscopic properties of ambient SSAs
and parametrizing phase changes for model applications.

Mineral particles, such as aluminosilicates and calcium carbonate, which are typically non-

hygroscopic, can become hygroscopic when they are internally mixed with SSAs or react with gaseous
species such as $NO_x$, $SO_2$, and organic acids in the presence of water vapor (Tang et al., 2016; Li et al.,
2014a). In fact, mineral dust and aged SSAs may exist as internal and/or external mixtures in the
atmosphere (Geng et al., 2014). For instance, Mg-silicate particle coagulated with SSA partially
increased in size only at the SSA region with increasing RH and was covered with an aqueous droplet
caused by the complete dissolution of the SSA part in high RH (Semeniuk et al., 2007). Additionally,
aluminosilicates coated with sulfur-bearing materials or internally mixed with sea salt particles can
absorb water, although the aluminosilicates remained as solid phases (Freney et al., 2010b).

In this study, we systematically investigated the hygroscopic behavior of ambient aerosols

collected on Jeju Island, Korea, together with their chemical compositions in various mixing states on
a single particle basis. Especially, the hygroscopic properties and chemical compositions of ambient
SSAs were examined and compared with multicomponent inorganic surrogate systems containing $Na^+$,
$Mg^{2+}$, $Cl^-$, and $NO_3^-$. The phase transitions were observed by monitoring the 2-D size changes of the
particles as a function of RH under optical microscopy and the hygroscopic curves and phase diagrams
were derived. To determine the chemical compositions of the individual ambient aerosols and the
spatial distribution of elements in the effloresced particles, we used scanning electron
microscopy/energy dispersive X-ray spectroscopy (SEM/EDX) and analyzed their X-ray spectra and
maps, respectively. Although ambient aerosols are complex in their compositions and hygroscopic
properties, the detailed elucidation of their hygroscopic behavior according to chemical compositions
of ambient aerosols can contribute to the ongoing efforts to improve our understanding of atmospheric
aerosols and their impacts on global climate.





## 2 Experimental Section

### 2.1 Samples

**Ambient aerosol particles**

Aerosol samples were collected on April 16 and 17, 2012, at the Gosan meteorological site (33.29°N, 126.16°E) located on the west coast of Jeju Island in South Korea (see Fig. 1). Ambient aerosols were loaded on TEM grids (200-mesh Cu coated with Formvar stabilized with carbon, Ted Pella, Inc.) mounted on stages 2 and 3 of a three-stage cascade $PM_{10}$ impactor (Dekati Ltd.) with aerodynamic cut-off diameters of 10–2.5 μm and 2.5–1.0 μm, respectively, at a flow rate of 10 L min⁻¹. Sampling durations for each stage were adjusted to collect an appropriate number of particles without overloading. Stage 2 particles, which were sized between 10-2.5 μm, were used to measure chemical composition and hygroscopic behavior in this study.

**Laboratory-generated (Na, Mg)(Cl, NO₃) mixture particles**

In the previous studies (Zhang et al., 2004; Gupta et al., 2015a; Gupta et al., 2015b), aerosols of NaCl, NaNO₃, Mg(NO₃)₂, NaCl-NaNO₃, and NaCl-MgCl₂ were extensively investigated. As $Na^+$, $Mg^{2+}$, $Cl^-$, and $NO_3^-$ are also major species of most ambient SSAs, we measured the hygroscopic behavior of NaCl-MgCl₂-NaNO₃-Mg(NO₃)₂ in this work to study the hygroscopic behavior of ambient SSAs that contain similar major elements. Pure solutions (1.0 M each) of NaCl (>99.9% purity, Aldrich), MgCl₂·6H₂O, and NaNO₃ (99.9% purity, Aldrich) were prepared using de-ionized water (18 MΩ, Millipore Direct-QTM). The pure solutions were then mixed to obtain mixture solutions with $[Cl^-]:[NO_3^-]$ = 3:1, 1:1, and 1:3 (i.e. $X_{(Na,Mg)Cl}$ = 0.75, 0.5, and 0.25) while the sea water ratio of $[Na^+]$ : $[Mg^{2+}]$ = 9:1 was maintained. A single jet atomizer (HCT4810) was used to generate aerosol particles from the mixture solutions on hydrophobic TEM grids. Herein, a notation system is used to represent aerosol particles of NaCl-MgCl₂-NaNO₃-Mg(NO₃)₂ as (Na, Mg)(Cl, NO₃).

### 2.2 Hygroscopic property measurements

The experimental setup for measuring hygroscopic behavior consists of three main components: (A) a see-through impactor, (B) an optical microscope, and (C) a humidity control system. The TEM grid with aerosol particles was attached to the impaction plate in the see-through impactor, and the RH was controlled by mixing dry and wet gaseous N₂ (99.999% purity) flows that were adjusted to obtain the desired RH in the range of ~5.0−92.0%. The humidity control system used wet N₂ gas obtained by



bubbling through deionized water reservoirs. The RH was monitored by a digital hygrometer (Testo
645) that was calibrated using a dew-point hygrometer (M2 Plus-RH, GE) to provide RH readings with
±0.5% reproducibility. A detailed discussion of the impactor and humidity-controlling system can be
found elsewhere (Li et al., 2021). The particles were continuously imaged in RH = 1% steps using a
digital camera (Canon EOS 5D, full frame, Canon EF f/3.5 L macro USM lens) mounted on an optical
microscope (Olympus, BX51M) during the humidification process (by increasing RH from ~5.0 to
92.0 %), followed by the dehydration process (by decreasing RH from ~92.0 to 5.0 %). The changes in
particle size with the variation of RH were monitored by measuring the particle areas in the optical
images to generate hygroscopic curves. Each humidity condition was sustained for at least 2 mins to
allow for sufficient time for water condensation or evaporation. The hygroscopic curves are represented
by the area ratio ($A/A_0$) as a function of RH, where the 2-D projected aerosol area at a given RH ($A$) is
divided by that before starting the humidification process ($A_0$). The images were processed using image
analysis software (Matrox, Inspector v9.0). The experiments were conducted at room temperature (T =
22 ± 1 ºC). Pure NaCl particles were used to verify the accuracy of the system with DRH = 75.5 (±
0.5) % and ERH = 46.3–47.6 %.

**2.3 Low-*Z* particle EPMA measurements using SEM-EDX**

The ambient aerosol particles were analyzed using low-*Z* particle EPMA measurements with a

Jeol JSM-6390 SEM equipped with an Oxford Link super atmospheric thin window (SATW) EDX
detector. The analysis was conducted both before and after the hygroscopic processes to determine the
morphology, chemical composition, and spatial distribution of the chemical elements (elemental maps).
The resolution of the detector was 133 eV for Mn Kα X-rays. Point mode and area mode X-ray spectra
and elemental maps of individual particles were recorded using Oxford INCA Energy software. An
accelerating voltage of 10 kV and beam current of 0.5 nA were used, and typical measurement durations
were 20 sec. for point mode, 1 min. for area mode, and 5-10 min. for elemental mapping.

The AXIL program was used to obtain the net X-ray intensities for chemical elements through

non-linear least-squares fitting of the spectra. From these intensities, the elemental concentrations of
individual particles were determined (Vekemans et al., 1994). For individual particles sitting on TEM
grids, C and O concentrations were determined using a Monte Carlo calculation technique to correct
for the interfering X-ray peaks of C and O emitted from the TEM grid, providing accurate quantification



results (Geng et al., 2010). A detailed explanation of the elemental quantification procedure can be
found elsewhere (Wu et al., 2019a).
**3 Results and Discussion**
**3.1 Chemical compositional analysis of individual ambient aerosol particles**
Firstly, low-$Z$ particle EPMA measurement was performed to find out fields on TEM grids with
well-separated particles based on their secondary electron images (SEIs, ~ 100 μm x 100 μm for a field)
before conducting the hygroscopic study and chemical compositional analysis of individual ambient
aerosols. The particles on the selected fields were monitored using an optical microscope at varying
RHs during the humidification and dehydration processes to study their hygroscopic behavior.
Subsequently, the effloresced particles were transferred back to SEM-EDX to obtain their SEIs and X-
ray spectra. In this study, a total of 39 particles on three fields were investigated, including 24 SSAs
with diverse mixing ratios of $Cl^-$ and $NO_3^-$ and 15 other particles such as six aluminosilicates, five Ca-
containing particles, two Fe-rich particles, an aged $SiO_2$, and a mixture particle of organic and
$(NH_4)_2SO_4$. The mole fraction of Cl as $X_{(Na,Mg)Cl}$ in the aged/reacted SSAs was calculated based on
$[Cl^-]/([Na^+]+2[Mg^{2+}])$ to determine the degree of Cl-depletion in the SSA particles. Fig. 2 shows the
SEI of the first field containing 16 particles, where the chemical species of each particle are indicated,
together with two exemplar X-ray spectra of aged SSAs #5 and #11 with $X_{(Na,Mg)Cl}$ = 0.75 and 0.23,
respectively. The elemental concentrations of all particles and their chemical species, determined by X-
ray spectral analysis, are listed in Table S1 of Supporting Information.
**3.2 Hygroscopic behavior of ambient aerosol particles**
The hygroscopic behavior of all 39 particles was investigated in detail, in conjunction with their
chemical compositional analysis. In Fig. 3, optical images obtained at different RHs during the
humidification and dehydration processes and the SEI after hygroscopic process for 16 particles on the
first field are shown. Except particles #2, #4, #7, #8, and #14, the rest are aged SSAs. Optical images
and the SEIs for particles on the second and third fields are provided in Figs. S1 and S2, respectively.
Particles were initially solid at RH = 5.3 % before the hygroscopic measurement, as shown in
Fig. 3A. During the humidification process, most of the SSAs showed partial deliquescence at RH =
63.8 % (Fig. 3C), regardless of their aging degree, indicating the realization of the MDRH. Upon further
increase of RH, SSAs underwent full deliquescence transitions at DRH = 65.4 – 73.5 %, which varied

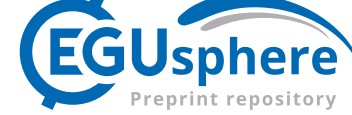

for each SSA, as shown in Figs. 3D and E. The SSA droplets showed hygroscopic growth when RH
increased further as shown in Fig. 3F. During the dehydration process, almost all SSAs exhibited
colored ring-type patterns (Figs. 3G and H) due to the diffraction of visible light typically observed in
the presence of organic surfactants on water, indicating the presence of considerable amounts of organic
species in aged SSAs. SSA droplet #5 crystallized at ERH = 50.5 % (Fig. 3G), while the others
effloresced over a lower range of ERH = 46.0 – 17.9%. The various DRHs and ERHs indicate different
chemical compositions of SSAs.

Aluminosilicates and Fe-rich particles did not exhibit any water uptake or changes as a function

of RH. Aged aluminosilicates, aged $SiO_2$, and some reacted Ca-containing particles showed modest
growth/shrinkage continuously due to the presence of amorphous $NO_3^-$ phases (Ahn et al., 2010). The
mixture particle of organic and ammonium sulfate experienced distinct deliquescence and efflorescence.
Detailed description of hygroscopic behavior of the ambient aerosol particles is given as follows.

**3.3 SSA particles**

The average atomic concentrations of C, Na, Cl, N, Mg, S, K, and Ca in SSAs are listed in Table

S1 with the values being 43.0(±7.2)%, 13.6(±4.6)%, 8.0(±6.6)%, 7.9(±2.3)%, 1.5(±0.4)%, 0.8(±0.3)%,
0.3(±0.1)%, and 0.3(±0.1)%, respectively. The elemental analysis of the SSAs indicates that they
consist primarily of $Na^+$, $Mg^{2+}$, $Cl^-$, $NO_3^-$, and organic species. As the SSAs become more aged or
reacted, their mole fractions of $Cl^-$ relative to $Na^+$ and $Mg^{2+}$ decrease, while their mole fractions of $NO_3^-$
increase. Although the aged/reacted SSAs are a complicated multi-component system, (Na,Mg)(Cl,
$NO_3$) mixture system is considered as an inorganic surrogate system for understanding their
hygroscopic behavior. The AIOMFAC model predicts the eutonic compositions of Na(Cl, $NO_3$) and
(Na, Mg)(Cl, $NO_3$) mixture systems as $X_{NaCl} = 0.38$ and $X_{(Na,Mg)Cl} = 0.46$, respectively. The mole
fraction of Cl in the eutonic compositions of the inorganic component of the aged SSAs would be
around 0.4. And thus, the SSAs can be classified as Cl-rich or Cl-depleted depending on their $X_{(Na,Mg)Cl}$
values being greater or less than 0.4, respectively.

**3.3.1 Cl-rich SSAs**

Fig. 4 displays the projected 2-dimensional (2-D) area ratio and optical images for Cl-rich SSAs

#5 and #19 (panels a and d, respectively), and the 2-D area ratio of humidification (panels b and e) and
dehydration (panels c and f) for the two SSAs as a function of RH. For SSAs #5 and #19 the mole



fraction of chloride, $X_{(Na,Mg)Cl}$ was calculated to be 0.75 and 0.72, respectively. In addition, the figure
also includes the (Na, Mg)(Cl, NO$_3$) particle with $X_{(Na,Mg)Cl}$ = 0.75 for comparison.

During the humidification process, SSAs #5 and #19 initially remained constant in size until

RH reached around 30%, after which they gradually increased in size. This behavior is also consistent
with the (Na, Mg)(Cl, NO$_3$) system (Figs. 4b and e), suggesting that MgCl$_2$·6H$_2$O (DRH = 33.3 %) or
Mg$^{2+}$-rich eutonic part may have undergone water absorption (Gupta et al., 2015a; Zieger et al., 2017).
However, the deliquescence transitions were not as distinct as those in particles with $X_{(Na,Mg)Cl}$ = 0.25
and 0.5 generated from (Na,Mg)(Cl,NO$_3$) mixture solutions (see Fig. S3 in the Supporting Information).
The size increase was followed by a shrinkage until RH = 60.5% due to structural rearrangement in the
remaining undissolved salt-mixture crystals. Structural rearrangements are commonly observed after
preliminary absorption of water at RHs just before the prompt deliquescence transition (Mikhailov et
al., 2009; Mikhailov et al., 2004; Ahn et al., 2010; Gupta et al., 2015a; Gupta et al., 2015b). At first
DRH = 63.8%, a partial droplet-like shape appeared in the particle morphology although there was no
significant change in the 2-D area ratio. Clear final deliquescence transitions were observed in both
SSAs #5 and #19 at DRHs = 73.5 % and 72.9 %, respectively, due to the dissolution of the remaining
solid NaCl moiety in these Cl-rich particles. The measured DRHs of SSAs #5 and #19 were closer to
the final DRHs calculated for the Na(Cl, NO$_3$) system using the AIOMFAC model (Fig. 6), and higher
than the calculated and measured ones in the (Na, Mg)(Cl, NO$_3$) particles (Figs. 4b and e and Fig. 6).
These observations for the Cl-rich SSAs suggest that most of the Mg$^{2+}$ salts (MgCl$_2$·6H$_2$O with DRH
= 33.3 %; Mg(NO$_3$)$_2$ with DRH = 52%) have already dissolved at low RHs or undergone complexation
with organic moieties (Eom et al., 2016), and the remaining NaCl and NaNO$_3$ moieties drove the
deliquescence transition. The hygroscopic growth of both SSAs #5 and #19 was much smaller than that
of the (Na, Mg)(Cl, NO$_3$) system (Figs. 4b and e) when RH was raised to 91.4%, indicating the presence
of a partitioning hydrophobic layer composed of organic surfactants that covered the aqueous salt
droplets and inhibited water uptake (Eom et al., 2016; Cochran et al., 2016; Bertram et al., 2018; Lee
et al., 2020).

During the dehydration process as RH decreased from ~91% to ~5%, both the SSA droplets

showed a continuous shrinkage in size before their efflorescence. However, the rate of shrinkage was
much smaller than that of pure inorganic surrogates, such as the (Na, Mg)(Cl, NO$_3$) system with
$X_{(Na,Mg)Cl}$ = 0.75 (Figs. 4c and f), indicating that the hydrophobic surfactant layers covering the aqueous
salt droplets potentially impeded the water evaporation. SSAs #5 and #19 showed one clear





efflorescence transition at RH = 50.5% and 45.0%, respectively (Figs. 4a and d). Interestingly, SSA #5
($X_{(Na,Mg)Cl}$ = 0.75) underwent a sharp decrease in size from RH = 50.9 – 50.5%, which is considerably
higher than the ERH range of ~45 – 47% for pure NaCl particles (Martin, 2000; Ahn et al., 2010; Eom
et al., 2014), and the first ERHs of either Na(Cl, NO$_3$) (Fig. 7) or (Na, Mg)(Cl, NO$_3$) (Figs. 4c and 7)
systems. On the other hand, SSA #19 ($X_{(Na,Mg)Cl}$ = 0.72) first showed a small decrease in size at RH =
63.8 – 60.8%, which is not a typical efflorescence transition, followed by a sharp decrease in size at
RH = 45.8 – 45.0%, which is on the lower side of the ERH range for pure NaCl and slightly higher
than the first ERHs of Na(Cl, NO$_3$) and (Na, Mg)(Cl, NO$_3$) systems (Figs. 4f and 7). The higher ERH
at RH = 50.5 % in SSA #5 indicates heterogeneous efflorescence of the NaCl moiety, while the ERH =
45.0% in SSA #19 suggests the homogeneous nucleation of NaCl moiety. Both SSA droplets showed
just one decisive efflorescence transition at their ERHs, indicating co-crystallization of most aqueous
inorganic salt moieties along with NaCl, pointing towards the likelihood that the SSAs with more
complicated chemical compositions than the inorganic surrogate aerosols contain chemicals which can
act as seeds, such as (Na, Ca)SO$_4$ crystals, for the complete crystallization at the efflorescence transition.
These observations are substantiated by the distribution of S and O at the center and edge of the NaCl
moiety in the X-ray maps obtained from the effloresced SSA particles #5 and #19, respectively (Fig.
S4) (Li et al., 2014b; Gupta et al., 2015b). The apparent (not so sharp) decrease in size of SSA #19 at
RH = 63.8 – 60.8% could be attributed to: (i) the sudden shrinkage of viscous organic moieties covering
the aqueous salt droplet; or (ii) inhibited or slow water loss, probably due to a kinetic barrier to
crystallization from amorphous/gel forming moieties such as MgSO$_4$/Mg(NO$_3$)$_2$/(CH$_3$SO$_3$)$_2$(Mg, Ca)
(Xiao et al., 2008; Zhao et al., 2006; Zhang et al., 2004; Liu and Laskin, 2009); or (iii) a phase transition,
such as heterogeneous efflorescence, had occurred, but the presence of viscous moieties (organics or
Mg$^{2+}$- organic complex) affected the relative 2-D size decrease on the TEM grid substrate.

**3.3.2 Equimolar and Cl-depleted SSAs**

Fig. 5 shows the plots of the projected 2-D area ratio and optical images for an equimolar SSA

#23 (Fig. 5a) and a Cl-depleted SSA #11 (Fig. 5d), and 2-D area ratio of humidification (panels b and
e) and dehydration (panels c and f) for the two SSAs as a function of RH. The calculated mole fractions
of chloride, $X_{(Na, Mg)Cl}$ for SSAs #23 and #11 are 0.52 and 0.23, respectively. The figure also includes
the (Na, Mg)(Cl, NO$_3$) particles with $X_{(Na, Mg)Cl}$ = 0.5 and 0.25 for comparison.




340  During the humidification process, both SSAs #23 and #11 remained relatively constant until
341 RH = ~50 %, unlike the (Na, Mg)(Cl, NO₃) surrogates, probably due to the decreased concentration of
342 MgCl₂·6H₂O. The particle size then began to shrink until RH = ~59%. Both SSAs exhibited two distinct
343 deliquescence transitions. Partial deliquescence transitions occurred at RH = ~63.8%, which is the
344 MDRH for the mixture of soluble moieties in the ambient SSAs and was reported for the first time.
345 Both SSAs exhibited two clear deliquescence transitions. The observed and AIOMFAC-calculated
346 MDRHs of the Na(Cl, NO₃) system are ~68%, while the AIOMFAC-calculated second MDRH for the
347 (Na, Mg)(Cl, NO₃) system is 66.5% after the dissolution of MgCl₂·6H₂O at the calculated first MDRH
348 of ~34%, as shown in Fig. 6. This indicates that the eutonic component in the ambient SSAs, with a
349 lower MDRH of 63.8%, is composed of NaCl, NaNO₃, Mg(NO₃)₂, and some other minor, less-soluble
350 moieties. The mutual deliquescence transition was more distinct in Cl-depleted SSAs than in Cl-rich
351 SSAs, suggesting that the eutonic component in the ambient SSAs is richer with other salts compared
352 to NaCl. As RH increased further, both SSAs #23 and #11 underwent final deliquescence transitions at
353 DRHs = 69.5%. The observed DRH for SSA #23 ($X_{(Na,Mg)Cl}$ = 0.52) is closer to the final DRH calculated
354 from AIOMFAC for pure NaCl moiety in the Na(Cl, NO₃) system (Fig. 6), and higher than the
355 AIOMFAC-calculated final DRH for pure NaCl moiety in the (Na, Mg)(Cl, NO₃) system (Figs. 5b and
356 6). The observed DRH for SSA #11 ($X_{(Na,Mg)Cl}$ = 0.23) is lower than the AIOMFAC-calculated final
357 DRH (Fig. 6) for pure NaNO₃ moiety in the Na(Cl, NO₃) system (Gupta et al., 2015b), and close to the
358 AIOMFAC-calculated final DRH for pure NaNO₃ moiety in the (Na, Mg)(Cl, NO₃) system (Figs. 5e
359 and 6). The observation of Cl-depleted SSAs during humidification suggests that a (Na, Mg)(Cl, NO₃)
360 dominant system drives the deliquescence transition. As RH increased further, the hygroscopic growth
361 of both SSAs #23 and #11 was stunted, in comparison to the inorganic multicomponent surrogates (Figs.
362 5b and e), likely due to the presence of hydrophobic surfactants covering the aqueous salt droplet.

363  During the dehydration process, both SSAs #23 and #11 showed slower rates of shrinkage
364 compared to the pure inorganic surrogates (Figs. 5c and f), suggesting inhibition of water evaporation
365 due to surface hydrophobic organic moieties. The diffraction patterns at the aqueous salt droplet-
366 organic surfactant interface were more prominent in the form of color and/or ring-like patterns for
367 equimolar and Cl-depleted SSAs (Figs. 5a and d), indicating that the hydrophobic organic film may
368 become thicker or the concentration of organic surfactants may increase with aging. During dehydration,
369 SSA #23, which is equimolar or slightly Cl-rich, showed one sharp transition at RH = 44.1-43.6% and
370 a gradual decrease in size thereafter until RH = 30.7%, which is not considered an efflorescence




transition (Fig. 5a). The distinct ERH of 43.6% observed for SSA #23 was higher than the first ERHs
of both Na(Cl, NO$_3$) and (Na, Mg)(Cl, NO$_3$) systems (Fig. 5c and 7), indicating possible homogeneous
efflorescence of NaCl along with other salts on crystalline seeds such as (Ca, Na)SO$_4$ (Pósfai et al.,
1995; Semeniuk et al., 2007; Wise et al., 2007), as shown in the X-ray maps in Fig. S5a. The remaining
metastable amorphous/gel type NO$_3^-$ moieties and MgSO$_4$·xH$_2$O may be responsible for the gradual
decrease in size after the efflorescence transition (Li et al., 2016).  Cl-depleted SSA #11 showed two
clear efflorescence transitions at RH = 44.1–39.6% and 24.6–23.9%, as shown in Fig. 5d. The first
ERH of 39.6% measured for SSA #11 was also higher than the ERHs for both Na(Cl, NO$_3$) and (Na,
Mg)(Cl, NO$_3$) systems, while the second ERH of 23.9% was close to and lower than the ERH range for
pure NaCl in (Na, Mg)(Cl, NO$_3$) and Na(Cl, NO$_3$) systems, respectively (Figs. 5f and 7). The
observations for Cl-depleted SSA #11 suggest that the first ERH was most probably due to the
heterogeneous crystallization of the dominant NaNO$_3$ and/or Mg(NO$_3$)$_2$ on mixed cation sulfate
crystalline seeds such as (Ca, Na)SO$_4$, while NaCl continued to homogeneously nucleate until the
second ERH, where it crystallized at the center/core of the particle (Woods et al., 2013), as shown in
the X-ray maps in Fig. S5b.

**3.3.3 Phase diagrams of ambient SSAs in correlation with Na(Cl, NO$_3$) and (Na, Mg)(Cl, NO$_3$)**
**surrogates systems**

The phase diagrams of ambient SSAs in correlation with Na(Cl, NO$_3$) and (Na, Mg)(Cl, NO$_3$)

surrogate systems can show the relationship between the observed deliquescence and efflorescence
behavior of the ambient SSAs and those of the simpler surrogate systems. The ambient SSAs have more
complex compositions, but the simpler surrogate systems can help to identify the dominant salts and
their behavior in the ambient SSAs. For example, the observation of Cl-depleted SSAs during
humidification suggests that a (Na, Mg)(Cl, NO$_3$) dominant system drives the deliquescence transition.
This information can be useful in understanding the hygroscopic properties and behavior of
atmospheric aerosols.

**3.3.3.1 Deliquescence phase diagram**

Fig. 6 shows the experimentally measured DRHs for the ambient SSA particles and those of the

(Na, Mg)(Cl, NO$_3$) and Na(Cl, NO$_3$) systems, calculated from the AIOMFAC model, plotted as a
function of the mole fraction of chloride, f($X_{(Na, Mg)Cl}$ or $X_{NaCl}$).





The first MDRH of the (Na, Mg)(Cl, NO$_3$) system, which was calculated from AIOMFAC, is
34.2%, while the MDRH measured in laboratory-generated (Na, Mg)(Cl, NO$_3$) particles is ~33.4% (Fig.
S3). These values are attributed to the dominant MgCl$_2$·6H$_2$O eutonic component (Gupta et al., 2015a).
In some Cl-rich SSAs, a gradual increase in size and a change in morphology were observed at RH =
~33%, indicating that they were in the partial aqueous phase (Figs. 4a and d). It is also possible that the
gradual water uptake observed in SSAs at low RHs is due to amorphous inorganic moieties such as
MgSO$_4$·xH$_2$O (Zhao et al., 2006; Xiao et al., 2008) and Mg(NO$_3$)$_2$·xH$_2$O (Zhang et al., 2004), and/or
water-soluble secondary organics such as carboxylate salts formed due to reactions of the (Na, Mg)Cl
species with dicarboxylic acids, which are ubiquitous in the marine boundary layer (Laskin et al., 2012;
Ghorai et al., 2014; Li et al., 2021). Wise et al. (2009) reported that ambient SSAs started changing in
morphology at 36(±15)% RH.
For the (Na, Mg)(Cl, NO$_3$) system, the eutonic composition is $X_{(Na,Mg)Cl} = 0.46$, with a second
MDRH of 66.5%, calculated from AIOMFAC, and a measured value of ~66.6(±0.4)% in the laboratory-
generated (Na, Mg)(Cl, NO$_3$) particles (Fig. S3). For the Na(Cl, NO$_3$) system, the eutonic composition
is $X_{NaCl} = 0.38$, with the MDRH of 67.9%, calculated from AIOMFAC, and a measured value of
~67.9(±0.3)% in the laboratory-generated particles (Gupta et al., 2015b). The observed MDRH of
63.8(±0.3)% for the ambient SSAs (Fig. 6) is slightly lower than those of the surrogate systems,
indicating that the eutonic component in the mutual deliquescence transition may be Na(Cl, NO$_3$)-rich,
with minor concentrations of other soluble salt moieties, as the MDRH of salt mixtures is generally
lower than individual salt DRHs (Wexler and Seinfeld, 1991). It is less likely that Na$_2$SO$_4$ or CH$_3$SO$_3$Na
are major components of the eutonic component in these ambient SSAs, as they have much higher
MDRHs, i.e., MDRHs = 84.0% for Na(Cl, SO$_4$) and 71% for Na(Cl, CH$_3$SO$_3$) (Chang and Lee, 2002;
Liu and Laskin, 2009; Liu et al., 2011). The lowest final DRH measured for SSA #1 with $X_{(Na,Mg)Cl} =$
0.33 is at RH = 65.4 % (Fig. 6), which is close to the observed MDRH of 63.8(±0.3)%, suggesting that
the eutonic component likely has a chloride mole fraction within the range of $X_{(Na,Mg)Cl} =$ ~0.30–0.40.
Interestingly, Wise et al. (2009) reported rounding in particle morphology at RH = 65(±4)%, which is
close to the observed MDRH but was not confirmed as such since only four ambient SSA particles were
studied.
The final DRHs in both (Na, Mg)(Cl, NO$_3$) and Na(Cl, NO$_3$) systems are solely determined by
the solid salt remaining after the mutual deliquescence of the eutonic component. Fig. 6 clearly shows
that for the Cl-rich SSA particles with $X_{(Na,Mg)Cl} > 0.40$, which contain more NaCl than the eutonic



composition, the final DRH values (~67.5-73.5%) approached the DRH of pure NaCl salt (~75.3% at
298 K) as the chloride concentration increased. Similarly, for the Cl-depleted particles with $X_{(Na,Mg)Cl}$ <
0.40, the final DRH values (~65.4-72.9%) approached that of pure $NaNO_3$ salt (~74% at 298 K) as the
chloride concentration decreased.

The chemical components of each phase in the ambient SSAs during the humidification process

are not well known, therefore, five possible single and/or mixed phases are notated as alphabets (P, Q,
R, S, and T) and the possible major chemical components in each phase (s = solid; aq = aqueous) are
listed as follows:
(i)  P-(s): all components are mixed in solid phase at RH < ~33−35% at all mole fractions of chloride.
(ii) Q-(s + aq): a mixed phase comprising possibly aqueous $MgCl_2·6H_2O$ dominant eutonic

components, and solid NaCl + (Na, Mg, Ca)($NO_3$, $SO_4$) + organics between RH = ~33% and

the first clear MDRH of ~63.8%.

(iii) R-(s + aq): a mixed phase comprising solid $NaNO_3$ + (Ca, Na)$SO_4$ and aqueous eutonic

components rich in Na(Cl, $NO_3$) + Mg($NO_3$, $SO_4$, organics) between RH = 63.8% and final

DRHs for $X_{(Na,Mg)Cl}$ < 0.40, i.e., Cl-depleted SSAs.

(iv) S-(s + aq): a mixed phase comprising solid NaCl + (Ca, Na)$SO_4$ and aqueous eutonic

components rich in Na(Cl ,$NO_3$) + Mg(Cl, $NO_3$, $SO_4$, organics) between the RH = 63.8% and

final DRHs for $X_{(Na,Mg)Cl}$ > 0.40, i.e., Cl-rich SSAs.

(v)  T-(aq): aqueous phase for most components including NaCl + $NaNO_3$ + Mg(Cl, $NO_3$, $SO_4$,

organics) above the measured final DRHs at all mole fractions of chloride, while (Ca, Na)$SO_4$

should remain in crystalline solid phase and does not take part in the deliquescence transitions

in the measured RH range.


**3.3.3.2 Efflorescence phase diagram**

The experimentally measured ERHs for the ambient SSA particles and those of (Na, Mg)(Cl,

$NO_3$) and Na(Cl, $NO_3$) systems are plotted as a function of the mole fraction of chloride, f($X_{(Na,Mg)Cl}$ or
$X_{NaCl}$) in Fig. 7.

The first ERH values decrease from 44.5% to 24.8% and from 47.1% to 20.2% with decreasing

mole fractions of chloride for $X_{(Na, Mg)Cl}$ = 0.75 to 0.25 in the (Na, Mg)(Cl, $NO_3$) and $X_{NaCl}$ = 0.9 to 0.1
in the Na(Cl, $NO_3$) systems, respectively (Gupta et al., 2015b; Woods et al., 2013). This suggests that





the first efflorescence transitions in the surrogate systems are solely due to the homogeneous nucleation
of NaCl for both Cl-rich and Cl-depleted particles, and that the amorphous (Na, Mg)NO$_3$ species cannot
undergo homogeneous crystallization even at high supersaturation (Kim et al., 2012; Zhang et al., 2004).
In the case of Cl-rich ($X_{(Na,Mg)Cl}$ > 0.40) ambient SSAs, the first ERHs systematically decreased ranging
from 50.5% to 33.0% due to the homogeneous nucleation of NaCl for most particles. However, for a
few particles, such as SSA #5 (Fig. 4a, ERH = 50.5%), NaCl underwent heterogeneous crystallization
on the mixed cation sulfate ((Ca, Na)SO$_4$) crystalline seeds, resulting in higher ERH values. On the
other hand, in Cl-depleted ($X_{(Na,Mg)Cl}$ < 0.40) SSA particles, the first ERH values ranged from 46.0% to
24.6% as a random set of values higher than the first ERHs in either surrogate system (Fig. 7), indicating
heterogeneous crystallization of the richer (Na, Mg)NO$_3$ moieties on the mixed cation sulfate
crystalline seeds for most particles.
The second/final ERH in the Na(Cl, NO$_3$) system was only observed for Cl-rich particles ($X_{NaCl}$
> 0.38) due to the mutual efflorescence of the eutonic component ($X_{NaCl}$ = 0.38) at MERH =
~30.0–35.5%, while no second ERH was recorded for Cl-depleted particles as NaNO$_3$ heterogeneously
crystallized simultaneously on the homogeneously nucleated NaCl seeds (Gupta et al., 2015b). For the
(Na, Mg)(Cl, NO$_3$), a second ERH = ~29.6–27.4% was only observed for $X_{(Na, Mg)Cl}$ = 0.5 among the
three compositions measured (Fig. S3), probably due to the stochastic heterogeneous crystallization of
the Na(Cl, NO$_3$)-rich eutonic moiety on the NaCl seed. On the other hand, the typical transitions from
30.5% to 17.9% for the second/final ERH in ambient SSAs were only observed for Cl-depleted particles
with decreasing chloride concentration, i.e. $X_{(Na, Mg)Cl}$ = ~0.33 to 0.15, indicating homogeneous
nucleation of NaCl followed by assumed simultaneous heterogeneous crystallization of remaining
aqueous salt moieties such as Mg(SO$_4$, NO$_3$).
In the laboratory-generated (Na, Mg)(Cl, NO$_3$) particles with different mole fractions of
chloride, i.e. $X_{(Na, Mg)Cl}$ = 0.25, 0.5, 0.75 (Fig. S3), and 1.0 (Gupta et al., 2015a), clear final ERHs or
MERHs were observed at low RHs ranging in 11.0–5.1% ($X_{(Na, Mg)Cl}$ = 0.25, 0.5, and 1.0) and 14.6–12.1%
($X_{(Na, Mg)Cl}$ = 0.75), probably due to the crystallization of the dominant eutonic component of
MgCl$_2$·4H$_2$O and MgCl$_2$·6H$_2$O, respectively. However, such low values of ERHs were not observed in
the ambient SSAs, possibly because the concentrations of MgCl$_2$·xH$_2$O were too small to be detected
by the optical microscopy-derived 2-D area ratio. On the other hand, the very gradual shrinkage





observed in ambient SSAs at low RHs may be due to the presence of amorphous $Mg(NO_3, SO_4)\cdot xH_2O$
moieties, which were present in some particles (e.g., SSAs #5, #19, and #23 in Figs. 4a, 4d, and 5a).

494   Considering the possibility of water content at low RHs during the dehydration process (Cziczo

et al., 1997; Tang et al., 1997; Gupta et al., 2015a), the ambient SSAs can be divided into five potential
phases, denoted as alphabets (P, Q, R, S, T), based on the presence of different chemical components
and their states at different relative humidities (RHs) and the major chemical components in each phase
(s = solid; aq = aqueous) are listed as follows:

499  (i) P-(aq): Almost all components, including $NaCl$, $NaNO_3$, and $Mg(Cl, NO_3, SO_4,$ organics), are

500   mixed in the aqueous phase at RH > ~55% for all mole fractions of chloride. The $(Ca, Na)SO_4$

501   should remain in the crystalline solid phase as it does not take part in the phase transitions within

502   the measured RH range.

503  (ii) Q-(s + aq): a mixed phase including solid $NaCl$ and other heterogeneously crystallized moieties

504   + aqueous nucleating species like $(Mg\cdot xH_2O)^{2+}$ and $Cl^-/NO_3^-/SO_4^{2-}$ at RHs < 50.5% in Cl-rich

505   SSAs ($X_{(Na, Mg)Cl} > 0.40$).

506  (iii) R-(s + aq): a mixed phase including heterogeneously crystallized $(Na, Mg)NO_3\cdot xH_2O$ on

507   crystalline $(Ca, Na)SO_4(xH_2O)$ seeds + aqueous $NaCl$ and other moieties, between the first

508   ERHs = 46.0–24.6% and final ERHs = 30.5–17.9% for Cl-depleted SSAs ($X_{(Na, Mg)Cl} < 0.40$).

509  (iv) S-(s + aq): a mixed phase including solid $(Na, Mg, Ca)(NO_3, SO_4)\cdot xH_2O$ and homogeneously

510   crystallized $NaCl$ + aqueous/amorphous $(Mg\cdot xH_2O)^{2+}$ and $NO_3^-/SO_4^{2-}$, and other minor species

511   below final ERHs = 30.5–17.9% for Cl-depleted SSAs ($X_{(Na, Mg)Cl} < 0.40$).

512  (v) T-(s): All components are mixed in solid phase at RH = ~14.6–5.0% for all mole fractions of

513   chloride. Amorphous or gel forming $Mg(NO_3, SO_4)\cdot xH_2O$ shows gradual water loss, while only

514   a small amount of $MgCl_2\cdot xH_2O$ is expected to crystallize.


**3.4 Non-SSA particles**

517   Generally, mineral particles such as aluminosilicates and calcium carbonate tend to be difficult

to absorb water and grow in size with increasing RH. However, these particles can become hygroscopic
after reaction with $NO_x$ and $SO_2$ in the presence of water and/or mixing with SSAs. Particle #14 shown
in Fig. S6 is a highly aged aluminosilicate that has mixed with an SSA moiety, probably with (Na,
Mg)(Cl, $NO_3$) and organic species as confirmed using X-ray spectrum. Clear deliquescence and



efflorescence transitions are observed in the gradual growth and shrinkage of particle #14 due mainly
to the SSA part, and the growth of the aged aluminosilicate is certainly smaller than the SSA at
maximum RH. Particle #20 shown in Fig. S7 is a reacted Ca-containing particle with nitrate showing
gradual change in size during humidification and dehydration processes, following hygroscopic
property reported before (Ahn et al., 2010). Particle #36 is a typical ammonium sulfate mixed with
organic species as shown in Fig. S8, and the major chemical components of particle #36 are C, N, O,
and S. The particle showed a first partial deliquescence transition at RH = 67.5%, which may be the
MDRH for the mixture of ammonium sulfate and organic species. Upon further increase in RH, the
particle absorbed more moisture and fully dissolved at RH = 77.1%, which is slightly lower than the
DRH of pure ammonium sulfate particles (Wu et al., 2019). During the dehydration process, particle
#36 showed a slower rate of shrinkage than ammonium sulfate particles, indicating inhibition of water
evaporation due to surface hydrophobic organic moieties, and effloresced at RH = 27.6%, which is
lower than the ERH of ammonium sulfate particles (Wu et al., 2019). Some particles, such as reacted
Ca-containing particles #17 and #25 and an aged $SiO_2$ particle #37, exhibit partial dissolution as shown
in Fig. S9 and S10. The 2-D area ratio of particle #17 at maximum RH is larger than that of particle
#25, indicating the existence of more hygroscopic components in particle #17. Particle #37 showed
gradual increase and decrease with partial dissolution only on the right side of particle, possibly due to
the reacted Ca-containing one with nitrate moiety. The observation of the hygroscopic behavior of
particles having partial growth suggests that a small content of hygroscopic chemical species can
control the hygroscopic behavior of particles with major non-hygroscopic species. On the other hand,
genuine aluminosilicate particles #29 and #32 (Fig. S11) and Fe-rich particles #2 and #7 (Fig. S12) did
not show hygroscopic growth with changing RH.

**4 Atmospheric Implications**

The investigation of the hygroscopic behavior of ambient SSAs is crucial for understanding the

atmospheric chemistry and physics of marine environments. Previous studies have recognized that
SSAs contain a wide range of inorganic sea salt and organic species (Schiffer et al., 2018), making it
difficult to assess their hygroscopicity. In this study, the hygroscopic behavior of SSAs was
systematically characterized and correlated with the role of inorganic salt moieties and enriched organic
material coating. It was found that SSAs partially dissolve at lower RHs than the inorganic surrogates,
including (Na, Mg)(Cl, NO₃), due to the coexistence with other soluble moieties such as water-soluble



secondary organics. This indicates that SSAs will be increasingly susceptible to trace gas species and
subsequent heterogeneous chemical reactions (Lee et al., 2020). The degree of chloride depletion was
also found to affect the hygroscopic behavior of aged SSAs. SSAs with a higher degree of chloride
depletion, i.e., higher aging degree, tend to exhibit multiple phase transitions with reducing RH,
retaining phase-separated core-shell mixing state, which impacts aerosol-radiation interactions (Sun et
al., 2018). The importance of considering these surrogate systems when modeling the hygroscopic
behavior of ambient SSAs is apparent, especially in the case of field observations where a wide range
of mixing states exist in particles. Even though there are still discrepancies regarding whether the
organic fraction can influence the hygroscopic growth of SSAs, this study demonstrated that organic
substances covering the sea salt moieties did suppress their hygroscopic growth. Further investigations,
including obtaining exact elemental and molecular compositions of the organic shells, are needed to
examine the aging of SSAs and quantify their uncertain effects on SSAs' hygroscopicity in
thermodynamic models.

In addition, other species involving mineral dust and anthropogenic particles can be transported

into marine environments and mixed with SSAs, which alter the particle compositions and in turn, their
hygroscopicity. Heterogeneous mixing or coating of SSAs onto less hygroscopic dust particles can
enhance their ability to interact with water vapor and reactive trace gases, leading to the formation of
new particles and increased CCN activity (Tang et al., 2016). This mixing can also affect the optical
properties and radiative forcing of atmospheric aerosols, as the scattering and absorption of solar
radiation by aerosols are dependent on their size, composition, and mixing state. Furthermore, the
impact of anthropogenic emissions on the hygroscopicity of marine aerosols is an important area of
research, as increased levels of atmospheric pollutants may enhance the aerosol-water interaction and
lead to changes in cloud properties and precipitation patterns (Su et al., 2022). Therefore, a
comprehensive understanding of the hygroscopic behavior of marine aerosols and their interactions
with other atmospheric constituents is necessary for accurately predicting their impacts on climate and
air quality.

**5 Conclusions**

The hygroscopic behavior of individual ambient aerosol particles collected at a coastal site of

Jeju Island, Korea, was investigated in correlation with their chemical compositions derived from X-
ray microanalysis. Specifically, we focused on the hygroscopic behavior of ambient aged SSAs and



their dependence on the extent of reaction between $Cl^-$ and $NO_3^-$ ions, estimated from the mixing ratios
of these ions using SEM-EDX. The phase transitions of the aged SSAs were found to be dominated by
inorganics involving Na(Cl, $NO_3$) and/or (Na, Mg)(Cl, $NO_3$) systems, with organic surfactant films
covering the droplets suppressing hygroscopic growth and shrinkage with changing RH. For Cl-rich
SSAs, two major transitions were observed during the humidification process, firstly at the MDRH and
secondly at a final DRH. During the dehydration process, Cl-rich SSAs showed single-stage
efflorescence. Cl-depleted SSAs showed two prompt deliquescence transitions during the
humidification process and stepwise transitions during the dehydration process, depending on their
chemical compositions.

The hygroscopic behavior of other particle types, including aged aluminosilicate, Ca-containing,

organic and ammonium sulfate mixture, and Fe-rich particles, was also observed. Aged mineral
particles showed varying degrees of size changes with changing RH, potentially due to the presence of
SSAs and/or $NO_3^-$ species resulting from coagulation and heterogeneous reactions, while non-reacted
mineral and Fe-rich particles did not exhibit significant size changes during the hygroscopic process.
The mixture particles of organic and ammonium sulfate displayed lower DRH and ERH values
compared to pure ammonium sulfate salt, indicating the impact of organic species on the hygroscopic
behavior of ammonium sulfate. While there have been some studies on the hygroscopic behavior of
ambient marine aerosols, this study is one of the first to systematically investigate their hygroscopic
behavior and to correlate it with their chemical compositions, providing better insights into their impact
on climate change and atmospheric chemistry.

**Data availability**
The data used in this study are available upon request; please contact Chul-Un Ro (curo@inha.ac.kr).

**Author contributions**
LW, HJE, HY, DG, and HRC designed the experiment. LW, HJE, HY, and HRC carried out the
measurements and/or analyzed the data. LW, HJE, HY, DG, HRC, PF, and CUR contributed
discussion of the data. LW, HJE, DG, and CUR drafted the paper.

**Competing interests**
The authors declare that they have no conflict of interest.



**Acknowledgements**

This study was supported by the National Research Foundation of Korea (NRF) grant funded by the Korean government (MSIT) (No. 2021R1A4A1032579 and No. 2021R1A2C2004240) and by the National Institute of Environmental Research (NIER) funded by the Ministry of Environment (MOE) of Korea (NIER-2021-03-03-007).

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





**Figure 1.** Location of Gosan sampling site on Jeju Island, South Korea. (Map Copyright © Google Earth)







**Figure 2.** (a) Secondary electron image (SEI) of the first field containing 16 particles and X-ray spectra of (b) a Cl-rich and (c) a Cl-depleted aged SSAs (particles #5 and #11, respectively). In this image, aged SSAs, Fe-containing particles, and aluminosilicates are denoted as "r-SS", "Fe-rich", and "AlSi", respectively.





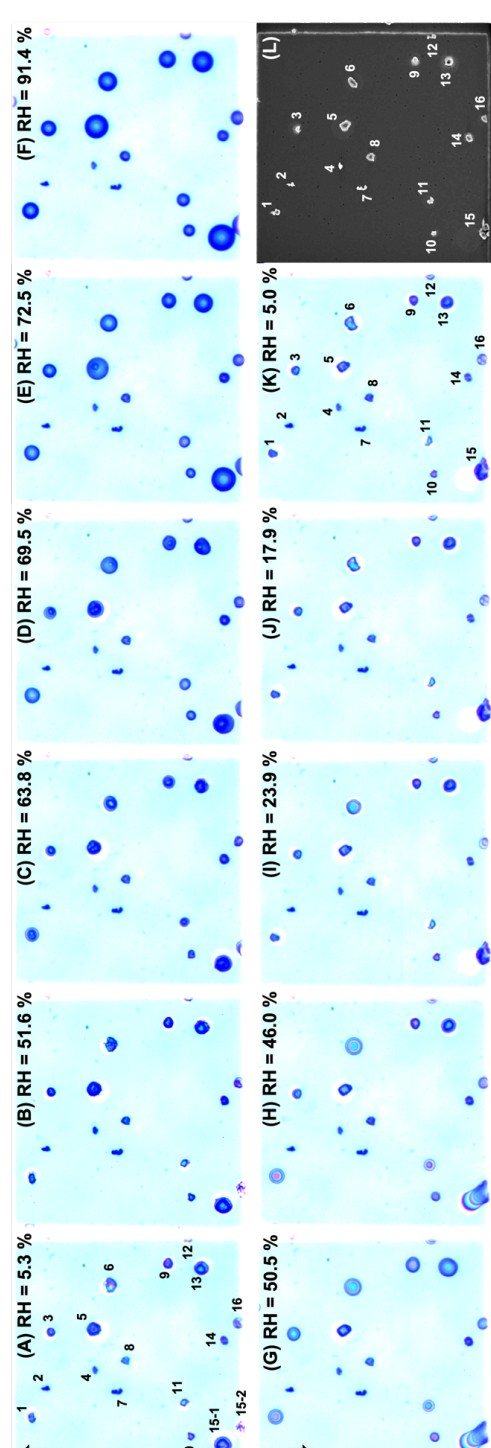

**Figure 3.** Optical images of the first field on TEM grid during humidifying (A-F, ↑) and dehydration (G-K, ↓) processes and the SEI of the same field (L).




**Figure 4.** 2-D area ratio plot and optical images of Cl-rich SSAs #5 and #19 (a and d), 2-D area ratio plots of humidification (b and e) and dehydration (c and f) for the SSAs and $(Na,Mg)(Cl,NO_3)$ particle as a function of RH.





**Figure 5.** 2-D area ratio plot and optical images of an equimolar SSA #23 and a Cl-depleted SSA #11 (a and d), 2-D area ratio plots of humidification (b and e) and dehydration (c and f) for the SSAs and $(Na,Mg)(Cl,NO_3)$ particles as a function of RH.



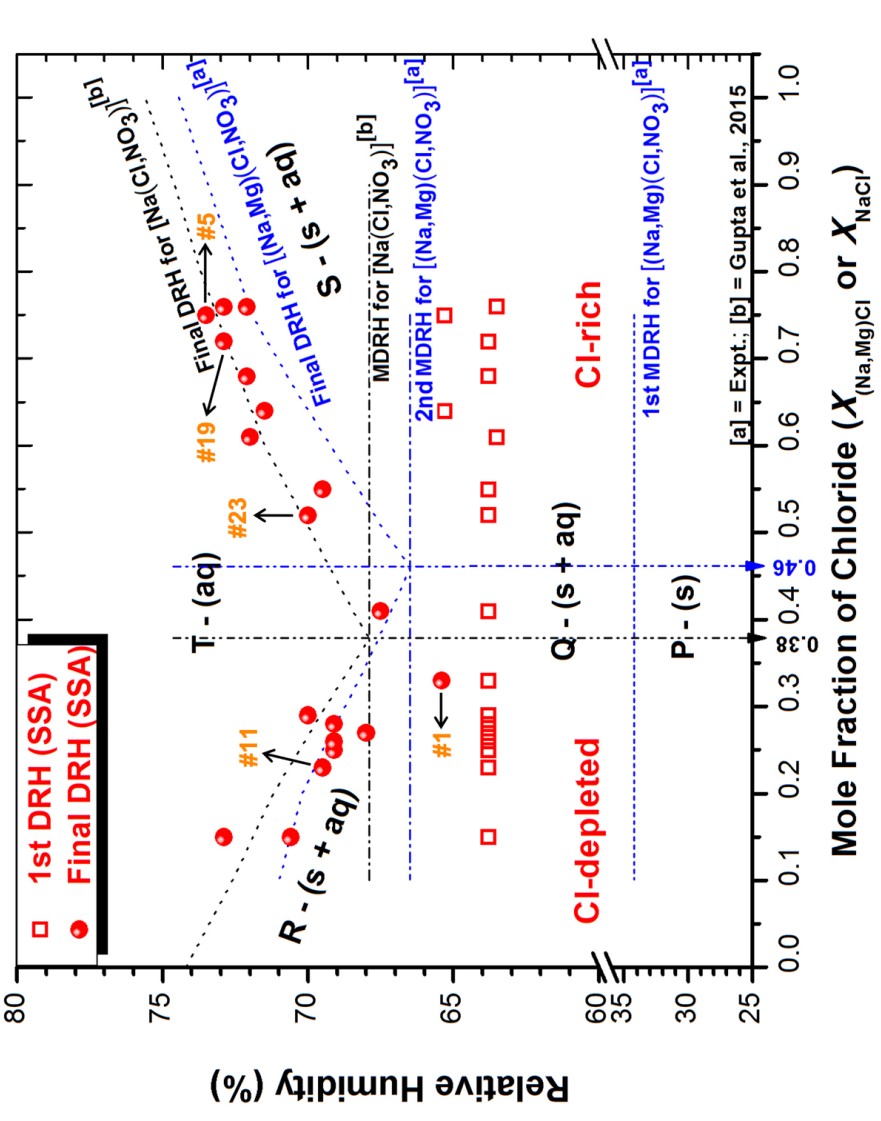

**Figure 6.** DRHs of ambient SSAs and (Na, Mg)(Cl, NO$_3$) and Na(Cl, NO$_3$) systems calculated from AIOMFAC plotted against the mole fraction of chloride [$X_{(Na, Mg)Cl}$ or $X_{NaCl}$].





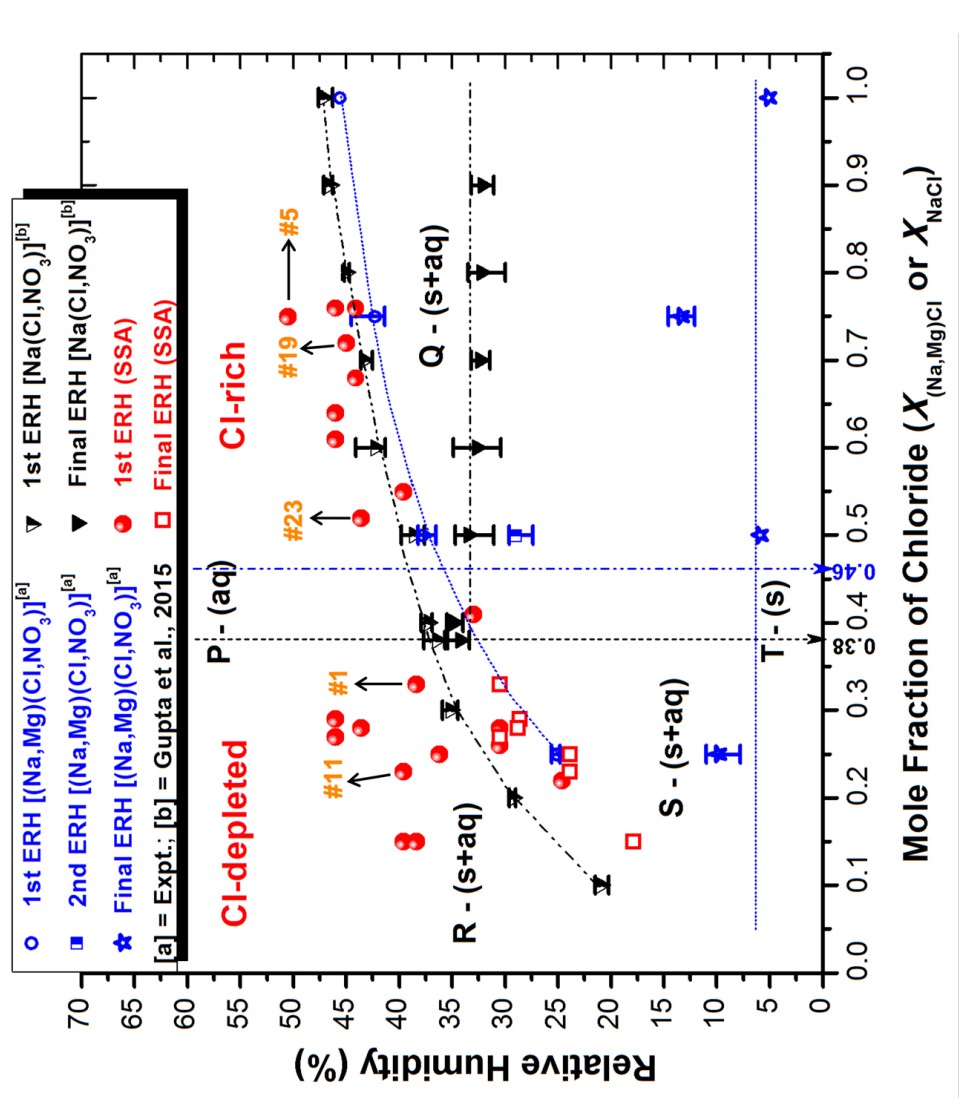

**Figure 7.** ERHs of ambient SSAs and those experimentally determined for (Na, Mg)(Cl, NO₃) and Na(Cl, NO₃) systems plotted against the mole fraction of chloride [$X_{(Na,Mg)Cl}$ or $X_{NaCl}$].
