# Peer review of "Chemical Composition-Dependent Hygroscopic Behavior of Individual Ambient Aerosol Particles Collected at a Coastal Site"

_EGUsphere, 2023_

## Referee Comment (RC1)

In their manuscript, "Chemical Composition-Dependent Hygroscopic Behavior of Individual Ambient Aerosol Particles Collected at a Coastal Site" the authors describe the deliquescence and efflorescence behaviour of ambient aerosol particles and correlate this with chemical composition. They also compare to a laboratory proxy system of known chemical composition and predict the deliquescence behaviour with AIOM-FAC calculations. Using all of this data, the authors generate phase diagrams for sea spray aerosol, having complex chemical composition, through the deliquescence and efflorescence processes. I believe this work provides novel insight to the deliquescence and efflorescence behaviour of complex mixtures representative of ambient aerosol. Below I have detailed a few points that should be addressed and a number of comments to help with the readability of the manuscript.

**Major Points**

1. There are a couple key, but specialised terms in this manuscript that have not been defined clearly, making it difficult for a non expert to follow. I suggest explicitly defining eutonic composition and mutual DRH/ERH.

2. The manuscript is missing a description of AIOMFAC calculations. How did you go from model output to DRH? Were these calculations done for the lab generated aerosol only or also every unique composition of sea spray?

3. Could the authors please add some details about how the ambient aerosol samples were stored before analysis. Is there potential for changes in their chemical composition or morphology between collection and measurement?

4. Could the authors please comment on the use of nitrate salts but not sulfate salts in their sea spray mimic - in the introduction sulfate was listed as a component of SSA but nitrate was not.

5. Could the authors comment on their choice to use the ratio of ions in sea water. Divalent cations are known to be enriched in in sea spray aerosol (Jayarathne et al. (2016). Enrichment of Saccharides and Divalent Cations in Sea Spray Aerosol During Two Phytoplankton Blooms. Environ. Sci. Technol., 50(11511–11520). https://doi.org/10.1021/acs.est.6b02988)

**Minor Points**

1. Line 115 - 119 - This statement is not clear to me. It probably needs to be broken into 2-3 sentences. Is this the case for all binary salt mixtures or just some?

2. Line 242 - what are these exception particles chemical compositions?

3. Line 255 - are there figures for these particles?

4. Does see-through mean transparent? Consider making this substitution in the manuscript.

5. Section 3.2 seems redundant with the methods and the following sections. What is the take away message from this section that is not provided in the later discussion of the specific particle types?

6. I think for clarity, it would be helpful to give the lab generated SSA (surrogate system) a specific name and use it throughout. It seems like sometimes they are called surrogate systems and sometimes just systems, and it becomes hard to follow if it is the lab proxy or some ambient aerosol that has that composition.

7. Section 3.3 could use an intro sentence that reminds the reader that SSA is a fraction of the ambient aerosol.

8. Line 301 - 304 & 305 - 309 & 365 - 368 was there evidence of organic in the elemental analysis as well?

9. Figure 3 scale bar for optical images? It is hard to tell what is 20 $\mu$m in the final panel. The whole field of view?

10. Fig 4/5 - are the lines just joining the points or do they represent something?

11. There are many references to the AIOMFAC calculated DRH, but they are not shown anywhere. Consider tabulating these for the SI.

12. The captions for Figures 6 and 7 need to be modified so that the reader can understand what is going on in the figure without having to go back through the main text. For example, the two vertical lines are not defined and the phases are not easy to understand without the definitions for P, S, R, Q & T.

13. Could the authors please check that all the citations made in the text have an entry in the reference section. I noticed Liu et al. 2011 is missing but did not do a thorough check.

---

## Author Comment (AC1)

**Reply to the comments of Anonymous Referee #1**

We thank the reviewer very much for the positive evaluation and valuable comments for our work. We appreciate thorough review and suggestions. Here, we provide a point-by-point response to the reviewer's specific comments. The final revision will be made later based on all reviewers' comments.

**Specific comments from Anonymous Referee #1 (comments are in italic)**

**Major Points**

1. *There are a couple of key, but specialised terms in this manuscript that have not been defined clearly, making it difficult for a non expert to follow. I suggest explicitly defining eutonic composition and mutual DRH/ERH.*

   **Response:** Thanks for the suggestion. Thermodynamic principles predict that stepwise phase transitions generally occur for two-component solid salt mixtures that can dissolve during the humidification process (Wexler, A. S. and Seinfeld, J. H.: Second-generation inorganic aerosol model, Atmos. Environ., 27A, 2731–2748, doi:10.1016/0960-1686(91)90203-J, 1991). When the mixing ratios of the two-component are at certain values, e.g. $X_{NaCl} = 0.38$ (or $X_{NaNO_3} = 0.62$) in $Na(Cl, NO_3)$ mixture, particles act like a single-salt with single phase transitions at the mutual deliquescence relative humidity (MDRH), where the mixture is considered as eutonic composition. For particles with rest mixing ratios ($X_{NaCl} > 0.38$ or $X_{NaNO_3} < 0.62$, termed as NaCl-rich; $X_{NaCl} < 0.38$ or $X_{NaNO_3} > 0.62$, termed as NaNO$_3$-rich), the first transition generally occurs at their MDRH, regardless of their mixing ratios; and the aqueous phase resulting from the partial deliquescence has the eutonic composition. The partially dissolved particles keep absorbing water with further increases in the RH and the residual solid component completely dissolves when the RH reaches their DRH, which depends on the richer salt moiety of the particle. During the dehydration process, the concentration of the richer salt moiety becomes dense and firstly crystallize at their efflorescence RH (ERH), and the aqueous phase of the eutonic composition effloresces at their mutual ERH (MERH) with further decreasing RHs.
   A more clear and readable description will be provided in the revised version.

2. *The manuscript is missing a description of AIOMFAC calculations. How did you go from model output to DRH? Were these calculations done for the lab generated aerosol only or also every unique composition of sea spray?*

**Response:** A description of AIOMFAC calculations will be provided in the revised version. These calculations were done for the proxy systems including (Na, Mg)(Cl, $NO_3$) and Na(Cl, $NO_3$) systems with various mixing ratios of the components.

3.  *Could the authors please add some details about how the ambient aerosol samples were stored before analysis. Is there potential for changes in their chemical composition or morphology between collection and measurement?*

**Response:** Our samples were put into the black plastic box and sealed with parafilm immediately after collection and stored into the refrigerator before measurements. The information will be added into the revised version.

4.  *Could the authors please comment on the use of nitrate salts but not sulfate salts in their sea spray mimic - in the introduction sulfate was listed as a component of SSA but nitrate was not.*

**Response:** Based on our low-*Z* particle EPMA results, the elemental concentration of N (7.9 (±2.3)%) was much higher than that of S (0.8(±0.3)%, line 262-264), and the elemental ratio of S and Na in SSAs collected in our samples is similar to that of sea water ratios ([S]/[Na] = 0.06, Haynes, W. M.: CRC Handbook of Chemistry and Physics, 96th Edition, CRC Press, Boca Raton, Florida, 14–18, 2015.), so that the SSAs were considered to contain more nitrate due to the rapid reaction with $NO_x$/$HNO_3$ instead of sulfate. In addition, sulfate salts generally have DRHs higher than 90%, which is not consistent with our hygroscopic observations as well. A more detailed explanation will be added in the revised version.

5.  *Could the authors comment on their choice to use the ratio of ions in sea water. Divalent cations are known to be enriched in in sea spray aerosol (Jayarathne et al. (2016). Enrichment of Saccharides and Divalent Cations in Sea Spray Aerosol During Two Phytoplankton Blooms. Environ. Sci. Technol., 50(11511–11520). https://doi.org/10.1021/acs.est.6b02988)*

**Response:** Please refer to our low-*Z* particle EPMA results. The elemental ratio of Na:Mg is around 9, which is also close to the seawater ratio (Haynes, 2015), so that we chose to maintain it. It is true that the divalent cations were observed to be enriched during phytoplankton blooms. However, it might not be encountered during our sampling. A explanation will be added into the revised version.

**Minor Points**

1. *Line 115 - 119 - This statement is not clear to me. It probably needs to be broken into 2-3 sentences. Is this the case for all binary salt mixtures or just some?*

   **Response:** As for the statement, please see the response to major point #1, and a more readable paragraph will be provided in the revised version. Theoretically, this is the case for binary salt mixtures in general.

2. *Line 242 - what are these exception particles chemical compositions?*

   **Response:** Please refer to the secondary electron image in Fig. 2, particles #2 and #7 are Fe-rich, particle #4 is aluminosilicate, and particles #8 and #14 are aged aluminosilicates. Their chemical compositions were provided in Table S1 in the supporting information.

3. *Line 255 - are there figures for these particles?*

   **Response:** Yes, please refer to Fig. S7-S12 in the supporting information.

4. *Does see-through mean transparent? Consider making this substitution in the manuscript.*

   **Response:** Yes, and the term was coined when we developed the system.

5. *Section 3.2 seems redundant with the methods and the following sections. What is the take away message from this section that is not provided in the later discussion of the specific particle types?*

   **Response:** Section 3.2 is a brief description of the hygroscopic behavior of different kinds of particles that we observed. This section will be simplified or deleted to avoid redundance.

6. *I think for clarity, it would be helpful to give the lab generated SSA (surrogate system) a specific name and use it throughout. It seems like sometimes they are called surrogate systems and sometimes just systems, and it becomes hard to follow if it is the lab proxy or some ambient aerosol that has that composition.*

   **Response:** Thanks very much to the suggestion of the reviewer, we will figure out a clear way to name these systems to avoid this confusion in the revised version.

7. *Section 3.3 could use an intro sentence that reminds the reader that SSA is a fraction of the ambient aerosol.*

   **Response:** The sentence will be added into the revised version.

8. *Line 301 - 304 & 305 - 309 & 365 - 368 was there evidence of organic in the elemental analysis as well?*

**Response:** Yes, the C contents, which represents organics, were clearly observed in the elemental analysis. Please refer to line 262-264, X-ray spectra in Fig.2, and the X-ray maps in Fig S4 and S5 in the supporting information.

9.  *Figure 3 scale bar for optical images? It is hard to tell what is 20 μm in the final panel. The whole field of view?*
    **Response:** More clear scale bars will be added both for optical images and SEI.

10. *Fig 4/5 - are the lines just joining the points or do they represent something?*
    **Response:** The lines just join the points for better demonstration of the phase transitions of particles.

11. *There are many references to the AIOMFAC calculated DRH, but they are not shown anywhere. Consider tabulating these for the SI.*
    **Response:** The AIOMFAC calculated DRHs are the results from thermodynamic predictions which are based on a thermodynamic equilibrium model.

12. *The captions for Figures 6 and 7 need to be modified so that the reader can understand what is going on in the figure without having to go back through the main text. For example, the two vertical lines are not defined and the phases are not easy to understand without the definitions for P, S, R, Q & T.*
    **Response:** Thank the reviewer for the suggestion, more readable figures will be provided in the revised version.

13. *Could the authors please check that all the citations made in the text have an entry in the reference section. I noticed Liu et al. 2011 is missing but did not do a thorough check.*
    **Response:** Thanks for the suggestion, the missing citation will be added, and all the citations will be carefully and thoroughly rechecked.

---

## Author Response (AR1)

**Reply to the comments of Anonymous Referees**

We thank the reviewers very much for the careful evaluation and valuable comments for our work. We revised the manuscript as much as possible respecting the reviewers' comments.

**Specific comments from Anonymous Referee #1 (comments are in italic)**

**Major Points**

1. *There are a couple key, but specialised terms in this manuscript that have not been defined clearly, making it difficult for a non expert to follow. I suggest explicitly defining eutonic composition and mutual DRH/ERH.*

**Response:** Thanks for the suggestion. Thermodynamic principles predict that stepwise phase transitions generally occur for two-component solid salt mixtures that can dissolve during the humidification process (Wexler, A. S. and Seinfeld, J. H.: Second-generation inorganic aerosol model, Atmos. Environ., 27A, 2731–2748, doi:10.1016/0960-1686(91)90203-J, 1991). When the mixing ratios of the two-component are at certain values, e.g. $X_{NaCl} = 0.38$ (or $X_{NaNO_3} = 0.62$) in Na(Cl, NO$_3$) mixture, particles act like a single-salt with single phase transitions at the mutual deliquescence relative humidity (MDRH), where the mixture is considered as eutonic composition.

For particles with other mixing ratios ($X_{NaCl} > 0.38$ or $X_{NaNO_3} < 0.62$, termed as NaCl-rich; $X_{NaCl} < 0.38$ or $X_{NaNO_3} > 0.62$, termed as NaNO$_3$-rich), the first transition generally occurs at their MDRH, regardless of their mixing ratios; and the aqueous phase resulting from the partial deliquescence has the eutonic composition. The partially dissolved particles keep absorbing water with further increases in the RH and the residual solid component completely dissolves when the RH reaches their DRH, which depends on the richer salt moiety of the particle. During the dehydration process, the concentration of the richer salt moiety becomes dense and firstly crystallize at their efflorescence RH (ERH), and the aqueous phase of the eutonic composition effloresces at their mutual ERH (MERH) with further decreasing RHs.

A more clear and readable description is provided in the revised version, please refer to line 115-127.

2. *The manuscript is missing a description of AIOMFAC calculations. How did you go from model output to DRH? Were these calculations done for the lab generated aerosol only or also every unique composition of sea spray?*

**Response:** A description of AIOMFAC calculations is provided in the revised version at line 132-137. These calculations were done for the proxy systems including $(Na, Mg)(Cl, NO_3)$ and $Na(Cl, NO_3)$ systems with various mixing ratios of the components.

3. *Could the authors please add some details about how the ambient aerosol samples were stored before analysis. Is there potential for changes in their chemical composition or morphology between collection and measurement?*

**Response:** The samples were put into black plastic boxes and sealed with parafilm immediately after the collection and these sealed samples were then stored in a refrigerator before the measurements (this paragraph is added at lines 179-180 in the revised version). We believe that this procedure minimizes the sample particle modification, if any.

4. *Could the authors please comment on the use of nitrate salts but not sulfate salts in their sea spray mimic - in the introduction sulfate was listed as a component of SSA but nitrate was not.*

**Response:** As shown in the manuscript, the elemental concentration of N, based on our low-$Z$ particle EPMA results, is much higher than that of S (7.9($\pm$2.3)% vs. 0.8($\pm$0.3)%). And the elemental ratio of S and Na in SSAs collected in our samples is similar to that of sea water ratios ([S]/[Na] = 0.06, Haynes, W. M.: CRC Handbook of Chemistry and Physics, 96th Edition, CRC Press, Boca Raton, Florida, 14–18, 2015.), so that the SSAs were considered to contain more nitrate due to the reaction with $NO_x$/$HNO_3$ instead of sulfate. In addition, sulfate salts generally have DRHs higher than 90%, which is not consistent with our hygroscopic observations as well. A more detailed explanation is added in the revised version at line 187 and 274-278.

5. *Could the authors comment on their choice to use the ratio of ions in sea water. Divalent cations are known to be enriched in sea spray aerosol (Jayarathne et al. (2016). Enrichment of Saccharides and Divalent Cations in Sea Spray Aerosol During Two Phytoplankton Blooms. Environ. Sci. Technol., 50(11511–11520). https://doi.org/10.1021/acs.est.6b02988)*

**Response:** Please refer to our low-$Z$ particle EPMA results. The elemental ratio of Na:Mg is around 9, which is also close to the seawater ratio (Haynes, 2015), so that we chose to maintain it. It is true that the divalent cations were observed to be enriched during phytoplankton blooms. However, it might not happen during our sampling. An explanation is added into the revised version at lines 192-194.

**Minor Points**

1. *Line 115 - 119 - This statement is not clear to me. It probably needs to be broken into 2-3 sentences. Is this the case for all binary salt mixtures or just some?*

   **Response:** As for the statement, please see the response to major point #1. Theoretically, this is the case for all binary inorganic salt mixtures.

2. *Line 242 - what are these exception particles chemical compositions?*

   **Response:** Please refer to the secondary electron image in Fig.2, where particles #2 and #7 are Fe-rich, particle #4 is aluminosilicate, and particles #8 and #14 are aged aluminosilicates. Their chemical compositions were provided in Table S1 in the supporting information.

3. *Line 255 - are there figures for these particles?*

   **Response:** Yes, please refer to Fig. 8 and S6-S11 in the supporting information.

4. *Does see-through mean transparent? Consider making this substitution in the manuscript.*

   **Response:** Yes, and the term was coined when we developed the system and has been used afterwards.

5. *Section 3.2 seems redundant with the methods and the following sections. What is the take away message from this section that is not provided in the later discussion of the specific particle types?*

   **Response:** Section 3.2 is a brief description of the hygroscopic behavior of different types of particles that we observed, including their ranges of DRHs and ERHs. This section is simplified and combined into section 3.1 (red marked in the revised version).

6. *I think for clarity, it would be helpful to give the lab generated SSA (surrogate system) a specific name and use it throughout. It seems like sometimes they are called surrogate systems and sometimes just systems, and it becomes hard to follow if it is the lab proxy or some ambient aerosol that has that composition.*

   **Response:** Thanks for the suggestion of the reviewer. The lab generated SSAs (inorganic surrogate system) in this study are termed as Na(Cl, NO$_3$) and (Na, Mg)(Cl, NO$_3$) systems. Please refer to lines 334, 374, 377, 405, 431-432, 476, and 485. The two parts including just "system" are changed as "composition". Please refer to lines 373 and 407.

7. *Section 3.3 could use an intro sentence that reminds the reader that SSA is a fraction of the ambient aerosol.*

   **Response:** The sentence is added into the revised version at line 280.

8.  *Line 301 - 304 & 305 - 309 & 365 - 368 was there evidence of organic in the elemental analysis as well?*

    **Response:** Yes. The C contents, which represents organics, were clearly observed in the elemental analysis. Please refer to lines 271-273, X-ray spectra in Fig. 2, and the X-ray maps in Fig S4 and S5 in the supporting information.

9.  *Figure 3 scale bar for optical images? It is hard to tell what is 20 μm in the final panel. The whole field of view?*

    **Response:** More clear scale bars are added both for optical images and SEI.

10. *Fig 4/5 - are the lines just joining the points or do they represent something?*

    **Response:** The lines just join the points for better illustration of the phase transitions of particles.

11. *There are many references to the AIOMFAC calculated DRH, but they are not shown anywhere. Consider tabulating these for the SI.*

    **Response:** The AIOMFAC-calculated DRHs are the results from thermodynamic predictions which are based on a thermodynamic equilibrium model.

12. *The captions for Figures 6 and 7 need to be modified so that the reader can understand what is going on in the figure without having to go back through the main text. For example, the two vertical lines are not defined and the phases are not easy to understand without the definitions for P, S, R, Q & T.*

**Response:** Thank the reviewer for the suggestion, more clear figure captions are provided in the revised version.

13. *Could the authors please check that all the citations made in the text have an entry in the reference section. I noticed Liu et al. 2011 is missing but did not do a thorough check.*

**Response:** Thanks very much to the reviewer for the suggestion, all the citations are carefully and thoroughly rechecked.

**Specific comments from Anonymous Referee #2 (comments are in italic)**

1. *I found that the paper was based on 39 particles collected in one sample. How could we ensure the representative of this sample?*

**Response:** Our main focus is not the characterization of the samples. Instead, we deeply investigated the hygroscopic behavior of ambient aerosols collected at a coastal site in terms of their chemical compositions.

2. *Line 95, an unnecessary parenthesis appeared in the sentence.*
**Response**: It's corrected.

3. *Figure 2, all the SSAs are classified as "aged SSA". Did the authors find any nascent SSA in their study?*

**Response:** Indeed, all the SSAs are more or less aged. The information is added at line 241.

4. Figures 6&7, the caption is too simple. At least the notations of five phases should be explained in the caption.

**Response:** Thank the reviewer for the suggestion. More comprehensive figure captions are provided in the revised version.

5. *Section 3.4, although nonSSA particles is not the main component in this paper, pure description without figure is not convenient for reading. It would be better if the authors could move one representative figure from SI into the text in this section.*

   **Response:** Fig. 8 for an aged aluminosilicate particle mixed with SSA is added in the revised version.

6. *The scale bar in Figure S2 was missing too.*
   **Response:** The scale bars are added.

7. *It is necessary to present SEM images of laboratory-generated (Na, Mg)(Cl, NO₃) mixture particles, maybe in Figure S3.*
   **Response:** SEM images are added.

8. *Figures S6-S12, what did the size in SEM image stand for? Scale bar or diameter?*

   **Response:** The size are diameters. The information is added into the figure captions.

[revised manuscript text omitted]